# Formation Tracking of UAV-UGV Systems Over Event-Triggered Communications

Yancheng Yan
*School of Automation Engineering*
*University of Electronic Science and Technology of China*
Chengdu 611731, China
yanchyan@126.com

Tieshan Li
*School of Automation Engineering*
*University of Electronic Science and Technology of China*
Chengdu 611731, China
tieshanli@126.com

Yue Long
*School of Automation Engineering*
*University of Electronic Science and Technology of China*
Chengdu 611731, China
longyue@uestc.edu.cn

Hanqing Yang
*School of Automation Engineering*
*University of Electronic Science and Technology of China*
Chengdu 611731, China
hqyang5517@uestc.edu.cn

Hongjing Liang
*School of Automation Engineering*
*University of Electronic Science and Technology of China*
Chengdu 611731, China
lianghongjing@uestc.edu.cn

*Abstract*—For unmanned aerial vehicle-unmanned ground vehicle (UAV-UGV) systems with disturbances, this paper investigates the formation tracking issue under event-triggered communications. A hierarchical control strategy is proposed, comprising a distributed event-triggered filter and a local controller. The former is responsible for estimating the desired position of the leader, while the latter employs a filter-based robust controller to ensure the formation tracking errors converge to a small residual set. The salient feature lies in the fact that the necessity for continuous communication among neighbors is obviated, and it has wider robustness to disturbances. Finally, simulations are conducted to validate the feasibility of the proposed strategy, showing accurate formation tracking with reduced communication overhead.

*Index Terms*—UAV-UGV systems, Event-triggered control, Formation control, Hierarchical control.

## I. INTRODUCTION

The distributed coordination control of multi-agent systems has recently attracted significant attention, spurred by the swift development of unmanned equipment and full automation. The goal is to bring all agents to a consensus through local interactions with neighboring agents. Applications include unmanned aerial vehicle formations, mobile robots, and unmanned underwater vehicles [1]–[3]. A typical application is the formation tracking control of unmanned aerial vehicle-unmanned ground vehicle (UAV-UGV) systems. For example, consider a combined surveillance-reconnaissance mission which involves two groups of unmanned vehicles (ground and aerial). In such scenario, UAV and UGV are deployed to collaboratively establish a secure passage through hostile territory. The foundation for accomplishing these tasks lies in achieving formation control of the UAV-UGV system. Notably, the UAV-UGV systems possess homogeneous dynamics along with specific heterogeneous dynamics. For the UAV-UGV systems, in recent literature, the formation problem has been effectively addressed [4]–[6]. To mention a few, in [6], a distributed adaptive formation control approach was developed for human-in-the-loop heterogeneous UAV-UGV systems based on a unified model.

Nonetheless, most existing control algorithms are designed with continuous communication among its neighbor nodes, inevitably leading to substantial consumption of communication resources. To mitigate this problem, event-triggered coordination control has become a major and widely researched topic in recent years. In [7], an event-triggered strategy was employed, and the results indicate that, compared to traditional real-time sampling strategies, the event-triggered strategy can reduce unnecessary communication. Subsequently, there has been significant advancement in the study of event-triggered control. For example, in [8], the implementation of event-triggered control alleviates not only the computational load but also controller update frequency. In [9], the edge-based event-triggered consensus problem was addressed based on the prescribed performance approach. However, the aforementioned control methods are based on event-triggered mechanisms in the controller channel, which inevitably require

This work was supported in part by the National Natural Science Foundation of China under Grant 51939001, Grant 62322307, Grant 62273072, and Grant 62203088, in part by the Natural Science Foundation of Sichuan Province under Grant 2022NSFSC0903. (Corresponding author: Yancheng Yan.)

real-time monitoring using neighbors' information, contrary to the original intention of the event-triggered mechanism. Even though literature [9] extended to self-triggered mechanisms to address this issue, the parameter values for these self-triggered mechanisms were rather conservative. Therefore, it is particularly important to design an event-triggered control strategy that does not rely on neighbors' information for real-time monitoring.

Drawing from the aforementioned observations, this paper investigates the formation tracking for UAV-UGV systems over event-triggered communications. The distributed event-triggered filter for each agent is developed to estimate the state of leader. Based on this filters, the local controller is designed to achieve the leader-follower formation tracking. The contributions are detailed below.

1) By employing an derivative operator and its power, an auxiliary observer is designed, along with an event-triggered mechanism to reduce the high communication demands between agents. Compared to the literature [8], [9], this approach eliminates the need for real-time monitoring of neighboring agents' states. Additionally, the order of the observer matches that of the UAV, preventing the issue of the virtual controller being non-differentiable.

2) The leader-follower formation tracking problem for heterogeneous UAV-UGV systems was addressed. Specifically, based on hierarchical control and backstepping techniques, utilizing filters and local controllers, a distributed formation control strategy was proposed, achieving formation tracking of the UAV-UGV system under unknown disturbances and limited communication resources.

## II. PRELIMINARIES

### A. UAV Dynamics

The $l$-th UAV model [4] is given as

$$
\begin{aligned}
\ddot{\xi}_{l,x} &= m_l^{-1}\big[f_l\big(c_{\nu_{l,1}}s_{\nu_{l,2}}c_{\nu_{l,3}} + s_{\nu_{l,1}}s_{\nu_{l,3}}\big)\big] \\
\ddot{\xi}_{l,y} &= m_l^{-1}\big[f_l\big(c_{\nu_{l,1}}s_{\nu_{l,2}}s_{\nu_{l,3}} - s_{\nu_{l,1}}s_{\nu_{l,3}}\big)\big] \\
\ddot{\xi}_{l,z} &= m_l^{-1}\big(f_l c_{\nu_{l,1}}c_{\nu_{l,2}} - m_l g\big) \\
\ddot{\nu}_{l,1} &= \dot{\nu}_{l,2}\nu_{l,3}\frac{I_{ly} - I_{lz}}{I_{lx}} + \frac{1}{I_{lx}}\big(u_{l,\nu_1} - I_{lr}\bar{q}_l\dot{\nu}_{l,2}\big) \qquad (1) \\
\ddot{\nu}_{l,2} &= \dot{\nu}_{l,1}\dot{\nu}_{l,3}\frac{I_{lz} - I_{lx}}{I_{ly}} + \frac{1}{I_{ly}}\big(u_{l,\nu_2} + I_{lr}\bar{q}_l\dot{\nu}_{l,1}\big) \\
\ddot{\nu}_{l,3} &= \dot{\nu}_{l,1}\dot{\nu}_{l,2}\frac{I_{lx} - I_{ly}}{I_{lz}} + \frac{1}{I_{lz}}u_{l,\nu_3}
\end{aligned}
$$

where $c_\star = \cos(\star)$ and $s_\star = \sin(\star)$; $\boldsymbol{\xi}_l = [\xi_{l,x}, \xi_{l,y}, \xi_{l,z}]^\top$ and $\boldsymbol{\nu}_l = [\nu_{l,1}, \nu_{l,2}, \nu_{l,3}]^\top$ represent the position and the attitude angle of $l$-th UAV; $I_{lx}$, $I_{ly}$, and $I_{lz}$ are the body inertia; $I_{lr}$ and $m_l$ are the the inertia and the mass; $\bar{q}_l$ represents the residual rotor angular, $g$ is the acceleration of gravity and $f_l$, $u_{l,\nu_1}$, $u_{l,\nu_2}$, and $u_{l,\nu_3}$ are the control input generated by four rotors.

The UAV's attitude subsystem can be stabilized independently, this study primarily addresses the trajectory tracking problem of UAV-UGV systems. The model for the $l$-th UAV is expressed as follows:

$$
\ddot{\boldsymbol{\xi}}_l = m_l^{-1}\boldsymbol{u}_l + \boldsymbol{\vartheta}_l(t) \qquad (2)
$$

where $\boldsymbol{u}_l = [f_l\big(c_{\nu_{l,1}}s_{\nu_{l,2}}c_{\nu_{l,3}} + s_{\nu_{l,1}}s_{\nu_{l,3}}\big), f_l\big(c_{\nu_{l,1}}s_{\nu_{l,2}}s_{\nu_{l,3}} - s_{\nu_{l,1}}s_{\nu_{l,3}}\big), f_l c_{\nu_{l,1}}c_{\nu_{l,2}} - m_l g]^\top$, $\boldsymbol{\vartheta}_l(t) = [\vartheta_l^x(t), \vartheta_l^y(t), \vartheta_l^z(t)]^\top$ is the disturbance.

### B. UGV Dynamics

The kinematics of $l$-th UGV [6] is modeled as

$$
\begin{aligned}
\dot{\bar{\boldsymbol{\xi}}}_l(t) &= \begin{bmatrix} c_{\theta_l(t)} \\ s_{\theta_l(t)} \end{bmatrix} v_l(t) \\
\dot{\theta}_l(t) &= \omega_l(t)
\end{aligned} \qquad (3)
$$

where $\bar{\boldsymbol{\xi}}_l(t) = [\bar{\xi}_{l,x}(t), \bar{\xi}_{l,y}(t)]^\top \in \mathbf{R}^2$, $\theta_l(t)$, $v_l(t)$, and $\omega_l(t)$ denote the position w.r.t. the global coordinate frame, the heading angle, linear and angular velocity control inputs of $l$-th UGV. To implement the control of UGVs, we define the position $\boldsymbol{\xi}_l(t) = [\xi_{x,l}(t), \xi_{y,l}(t)]^\top \in \mathbf{R}^2$, which is shifted a non-zero distance $p_l > 0$ from $\bar{\boldsymbol{\xi}}_l(t)$ along direction $\theta_l(t)$. The position $\boldsymbol{\xi}_l(t)$ is described by $\boldsymbol{\xi}_l(t) = \bar{\boldsymbol{\xi}}_l(t) + p_l[c_{\theta_l(t)}, s_{\theta_l(t)}]^\top$, whose dynamics is modeled as

$$
\begin{aligned}
\dot{\boldsymbol{\xi}}_l(t) &= \begin{bmatrix} c_{\theta_l} & -p_l s_{\theta_l} \\ s_{\theta_l} & p_l c_{\theta_l} \end{bmatrix} \begin{bmatrix} v_l \\ \omega_l \end{bmatrix} \\
&= G_l \boldsymbol{u}_l
\end{aligned} \qquad (4)
$$

with a non-singular matrix $G_l = [c_{\theta_l}, -p_l s_{\theta_l}; s_{\theta_l}, p_l c_{\theta_l}]$ and input $\boldsymbol{u}_l = [v_l, \omega_l]^\top$. Consider the disturbance, the $l$-th UGV model is rewritten as follows:

$$
\dot{\boldsymbol{\xi}}_l = G_l \boldsymbol{u}_l + \boldsymbol{\vartheta}_l(t) \qquad (5)
$$

where $\boldsymbol{\vartheta}_l(t) = [\vartheta_l^v(t), \vartheta_l^\omega(t)]^\top$ is the disturbance.

### C. Graph Theory

The following table shows the relevant definitions used in the UAV-UGV system.

| Variables | : | Interpretations |
|---|---|---|
| $\mathbb{V} = \{1, \dots, N\}$ | : | the set of all UAVs and UGVs; |
| $\mathbb{F}_1, \mathbb{F}_2$ | : | index set of UAVs and index set of UGVs |
| $\mathbb{E} = \{(l, i) \mid l, i \in \mathbb{V}\}$ | : | the edge set; |
| $\mathbb{G} = \{\mathbb{V}, \mathbb{E}\}$ | : | a directed graph depicted the communication of the UAV-UGV system; |
| $\mathbb{A} = [a_{l,i}]_{N \times N}$ | : | the adjacency matrix with $a_{l,i} = 1$ if $(i, l) \in \mathbb{E}$ and $a_{l,i} = 0$ if $(i, l) \notin \mathbb{E}$; |
| $\mathbb{N}_l$ | : | $= \{i \in \mathbb{V} \mid (i, l) \in \mathbb{E}\}$, the set of neighbor agents; |
| $\mathbb{L} = \mathbb{D} - \mathbb{A}$ | : | the Laplacian matrix, where $\mathbb{D} = \text{diag}(\sum_{l \in \mathbb{N}_1} a_{1,l}, \dots, \sum_{l \in \mathbb{N}_N} a_{N,l})$; |
| $\mathbb{G}_0 = (\mathbb{V}_0, \mathbb{E}_0)$ | : | the augmented graph describing one leader and the UAV-UGV system; |
| $\mathbb{V}_0 = \{L, 1, \dots, N\}$ | : | the node set with $L$ being the leader; |
| $\mathbb{E}_0 = \{(l, i) \mid i, l \in \mathbb{V}_0\}$ | : | the edge set of $\mathbb{G}_0$; |
| $\mathbb{B} = \text{diag}(b_1, \dots, b_N)$ | : | the communication between the leader and the $l$-th agent with $b_l = 1$ if $(L, l) \in \mathbb{E}_0$ and $b_l = 0$ if $(L, l) \notin \mathbb{E}_0$. |

### D. Control Object

For the UAV-UGV system and the leader, design a control scheme such that
(1) all signals in the closed-loop are bounded;
(2) the formation tracking error satisfies:

$$\|\boldsymbol{\xi}(t) - \mathbf{1}_N \otimes \boldsymbol{\xi}_0(t) - \boldsymbol{o}\| < \rho, \qquad (6)$$

where $\xi_0$ is the leader output, $\rho > 0$ is an arbitrarily adjustable constant, $\mathbf{1}_N = [\underbrace{1, \ldots, 1}_{N}]^\top$, $\boldsymbol{\xi}(t) = [\boldsymbol{\xi}_{l_1}(t), l_1 \in \mathbb{F}_1, l_2 \in \mathbb{F}_2]^\top$, and $\boldsymbol{o} = [\boldsymbol{o}_1^\top, \ldots, \boldsymbol{o}_N^\top]^\top$ with $\boldsymbol{o}_l \in \mathbf{R}^3$ being the relative position deviation between the $l$-th agent's position and the leader's position.

The assumptions and lemmas are needed.

*Assumption 1:* The leader's output $\boldsymbol{\xi}_0$ and its derivatives $\dot{\boldsymbol{\xi}}_0$ and $\ddot{\boldsymbol{\xi}}_0$ are bounded and continuous.

*Assumption 2:* The disturbances are bounded, i.e., $\|\boldsymbol{\vartheta}_l(t)\| \leq \bar{\vartheta}_l$, where $\bar{\vartheta}_l > 0$ is a unknown constant.

*Assumption 3:* It is assumed that the graph $\mathbb{G}$ contains a spanning tree, and the root node is able to receive information from the leader.

*Remark 1:* Assumptions 1–3 are standard conditions for formation control design.

*Lemma 1 ( [10]):* For a nonsingular $M$-matrix $\mathbb{H} \in \mathbb{R}^{N \times N}$, there exists a matrix $\mathbb{P} > 0$ such that $\mathbb{Q} = \mathbb{P}\mathbb{H} + \mathbb{H}^\top \mathbb{P} > 0$.

## III. MAIN RESULTS

### A. Auxiliary Filters and Event-Triggering Mechanism

In order to implement event-triggered communication between agents, the following filters are designed:

$$\begin{cases} \dot{\boldsymbol{v}}_{l,1} = \boldsymbol{v}_{l,2} \\ \dot{\boldsymbol{v}}_{l,2} = \bar{\boldsymbol{v}}_l \end{cases} \qquad (7)$$

where $\bar{\boldsymbol{v}}_l \in \mathbf{R}^3$ is the filter input, for $l \in \mathbb{F}_1 \cup \mathbb{F}_2$, and

$$\begin{cases} \boldsymbol{v}_{0,1} = \boldsymbol{\xi}_0 \\ \boldsymbol{v}_{0,2} = \dot{\boldsymbol{\xi}}_0 \end{cases} \qquad (8)$$

for the leader. Define the formation error and the auxiliary error as $\boldsymbol{e}_l = \boldsymbol{v}_{l,1} - \boldsymbol{\xi}_0 - \boldsymbol{o}_l$ and $\bar{\boldsymbol{e}}_l = \boldsymbol{e}_l + \dot{\boldsymbol{e}}_l$. Taking its derivative yields

$$\begin{aligned} \dot{\bar{\boldsymbol{e}}}_l &= \dot{\boldsymbol{e}}_l + \ddot{\boldsymbol{e}}_l \\ &= \dot{\boldsymbol{v}}_{l,1} - \dot{\boldsymbol{\xi}}_0 + \bar{\boldsymbol{v}}_l - \ddot{\boldsymbol{\xi}}_0 \\ &= \bar{\boldsymbol{v}}_l - (\dot{\boldsymbol{\xi}}_0 + \ddot{\boldsymbol{\xi}}_0) + \boldsymbol{v}_{l,2} \end{aligned} \qquad (9)$$

In order to alleviate the communication pressure on the UAV-UGV system, the following event-triggered mechanism is established

$$t_{k,\mu+1} = \inf \left\{ t > t_{k,\mu} \Big| \|\tilde{\boldsymbol{h}}_k(t)\| \geq \frac{\psi_k}{c_v} \right\} \qquad (10)$$

if $k \in \mathbb{V}_0$ is a neighbour of another agent, where $t_{k,\mu+1}$ and $t_{k,\mu}$ represent the next trigger moment and the current trigger moment, where $\mu = 0, 1, 2, \ldots$, and $t_{k,0} = 0$; $\psi_k, c_v > 0$ are designed parameters; $\tilde{\boldsymbol{h}}_k(t) = \boldsymbol{h}_k(t) - \boldsymbol{h}_k(t_{k,\mu})$ is state

measurement error with $\boldsymbol{h}_k(t) = \boldsymbol{v}_{k,1} + \boldsymbol{v}_{k,2}$. Then we define the filter input as

$$\bar{\boldsymbol{v}}_l = -c_v \sum_{k \in \mathbb{N}_l} a_{lk}(\boldsymbol{h}_l(t) - \boldsymbol{h}_k(t_{k,\mu}) - \boldsymbol{o}_{lk}) - \boldsymbol{v}_{l,2} \qquad (11)$$

where $\boldsymbol{o}_{l,k} = \boldsymbol{o}_l - \boldsymbol{o}_k$.

*Remark 2:* According to equation (10), it can be observed that $\boldsymbol{h}_k(t)$ is only related to its own state and does not utilize information from neighboring agents. This avoids real-time monitoring of neighboring agents, effectively alleviating communication pressure between agents.

*Lemma 2:* As designed in (11) and (15), the auxiliary filters ensure the boundedness of $\boldsymbol{v}_{l_1,1}, \boldsymbol{v}_{l_1,2}$, and $\bar{\boldsymbol{v}}_l$ for $l_1 \in \mathbb{V}_0$ and $l \in \mathbb{V}$, and they compel $\boldsymbol{v}_{l,1}$ to follow $\boldsymbol{\xi}_0 + \boldsymbol{o}_l$, resulting in the formation error $\boldsymbol{e}_l = \boldsymbol{v}_{l,1} - \boldsymbol{\xi}_0 - \boldsymbol{o}_l$ converging to a small residual set. Furthermore, the Zeno behavior is strictly avoided.

*Proof:* Define the error as

$$\boldsymbol{\varepsilon}_l = \sum_{k \in \mathbb{N}_l} a_{lk}(\boldsymbol{v}_{l,1} - \boldsymbol{v}_{k,1} - \boldsymbol{o}_{lk}) \qquad (12)$$

whose compact form is

$$\boldsymbol{\varepsilon} = (\mathbb{H} \otimes I_3)(\boldsymbol{v}_1 - \mathbf{1}_N \otimes \boldsymbol{\xi}_0 - \boldsymbol{o}) = (\mathbb{H} \otimes I_3)\boldsymbol{e} \qquad (13)$$

where $\boldsymbol{\varepsilon} = [\varepsilon_1^\top, \ldots, \varepsilon_N^\top]^\top$, $\boldsymbol{v}_1 = [v_{1,1}^\top, \ldots, v_{N,1}^\top]^\top$, $\boldsymbol{o} = [o_1^\top, \ldots, o_N^\top]^\top$, and $\boldsymbol{e} = [e_1^\top, \ldots, e_N^\top]^\top$. Additionally, we define

$$\begin{aligned} \bar{\boldsymbol{\varepsilon}}_l &= \boldsymbol{\varepsilon}_l + \dot{\boldsymbol{\varepsilon}}_l \\ &= \sum_{k \in \mathbb{N}_l} a_{lk}(\boldsymbol{v}_{l,1} + \boldsymbol{v}_{l,2} - \boldsymbol{o}_{lk} - (\boldsymbol{v}_{k,1} + \boldsymbol{v}_{k,2})) \\ &= \sum_{k \in \mathbb{N}_l} a_{lk}(\boldsymbol{h}_l - \boldsymbol{h}_k - \boldsymbol{o}_{lk}) \end{aligned} \qquad (14)$$

whose compact form is

$$\bar{\boldsymbol{\varepsilon}} = (\mathbb{H} \otimes I_3)\bar{\boldsymbol{e}} \qquad (15)$$

where $\bar{\boldsymbol{\varepsilon}} = [\bar{\varepsilon}_1^\top, \ldots, \bar{\varepsilon}_N^\top]^\top$ and $\bar{\boldsymbol{e}} = [\bar{e}_1^\top, \ldots, \bar{e}_N^\top]^\top$. The derivative of $\bar{\boldsymbol{e}}_l$ is

$$\begin{aligned} \dot{\bar{\boldsymbol{e}}}_l &= \bar{\boldsymbol{v}}_l - (\dot{\boldsymbol{\xi}}_0 + \ddot{\boldsymbol{\xi}}_0) + \boldsymbol{v}_{l,2} \\ &= -c_v \sum_{k \in \mathbb{N}_l} a_{lk}(\boldsymbol{h}_l(t) - \boldsymbol{h}_k(t_{k,\mu}) - \boldsymbol{o}_{lk}) - (\dot{\boldsymbol{\xi}}_0 + \ddot{\boldsymbol{\xi}}_0) \end{aligned} \qquad (16)$$

According to $\tilde{\boldsymbol{h}}_k(t) = \boldsymbol{h}_k(t) - \boldsymbol{h}_k(t_{k,\mu})$, we have

$$\begin{aligned} \dot{\bar{\boldsymbol{e}}}_l &= -c_v \sum_{k \in \mathbb{N}_l} a_{lk}(\boldsymbol{h}_l(t) - \boldsymbol{h}_k(t) - \boldsymbol{o}_{lk}) - c_v \sum_{k \in \mathbb{N}_l} a_{lk}(\tilde{\boldsymbol{h}}_k(t)) \\ &\quad - (\dot{\boldsymbol{\xi}}_0 + \ddot{\boldsymbol{\xi}}_0) \\ &\leq -c_v \bar{\boldsymbol{e}}_l + c_v \sum_{k \in \mathbb{N}_l} a_{lk}\|\tilde{\boldsymbol{h}}_k(t)\| + \|\dot{\boldsymbol{\xi}}_0 + \ddot{\boldsymbol{\xi}}_0\| \end{aligned} \qquad (17)$$

It follows from (10) that the inequality $\|\tilde{\boldsymbol{h}}_k(t)\| \leq \frac{\psi_k}{c_v}$ is hold, then

$$\dot{\bar{\boldsymbol{e}}}_l \leq -c_v \bar{\boldsymbol{e}}_l + \sum_{k \in \mathbb{N}_l} a_{lk}\psi_k + \|\dot{\boldsymbol{\xi}}_0 + \ddot{\boldsymbol{\xi}}_0\| \qquad (18)$$

According to the Assumption 1, there exists a scalar vector $\phi_l > 0$ such that $\sum_{k\in\mathbb{N}_l} a_{lk}\psi_k + \|\dot{\boldsymbol{\xi}}_0 + \ddot{\boldsymbol{\xi}}_0\| \le \phi_l$. Thus, it yields

$$\dot{e}_l \le -c_v \bar{e}_l + \phi_l \tag{19}$$

Define the Lyapunov function as $V_o = \frac{1}{2}\bar{\boldsymbol{\varepsilon}}^\top (\mathbb{P} \otimes I_3)\bar{\boldsymbol{\varepsilon}}$ where $\mathbb{P} > 0$ is given in Lemma 1 as the Assumption 3 holds, taking its derivative along with (15) and (19) yields

$$\begin{aligned}
\dot{V}_o &= \bar{\boldsymbol{\varepsilon}}^\top (\mathbb{PH} \otimes I_3)\dot{\bar{\boldsymbol{e}}} \\
&\le -c_v \bar{\boldsymbol{\varepsilon}}^\top (\mathbb{PH} \otimes I_3)\bar{\boldsymbol{\varepsilon}} + \bar{\boldsymbol{\varepsilon}}^\top (\mathbb{PH} \otimes I_3)\boldsymbol{\phi}
\end{aligned} \tag{20}$$

where $\boldsymbol{\phi} = [\phi_1^\top, \ldots, \phi_N^\top]^\top$. By using the Lemma 1, one has

$$\dot{V}_o \le -\frac{c_v}{2}\underline{\lambda}(\mathbb{Q})\|\bar{\varepsilon}\|^2 + \bar{\boldsymbol{\varepsilon}}^\top (\mathbb{PH} \otimes I_3)\boldsymbol{\phi} \tag{21}$$

Using the Young's inequality, it yields

$$\bar{\boldsymbol{\varepsilon}}^\top (\mathbb{PH} \otimes I_3)\boldsymbol{\phi} \le \frac{1}{4}c_v\underline{\lambda}(\mathbb{Q})\|\bar{\varepsilon}\|^2 + \frac{\|(\mathbb{PH} \otimes I_3)\boldsymbol{\phi}\|^2}{c_v\underline{\lambda}(\mathbb{Q})} \tag{22}$$

Hence,

$$\begin{aligned}
\dot{V}_o &\le -\frac{c_v}{4}\underline{\lambda}(\mathbb{Q})\|\bar{\varepsilon}\|^2 + \frac{\|(\mathbb{PH} \otimes I_3)\boldsymbol{\phi}\|^2}{c_v\underline{\lambda}(\mathbb{Q})} \\
&\le -\frac{c_v\underline{\lambda}(\mathbb{Q})}{2\bar{\lambda}(\mathbb{P})}V_o + \frac{\|(\mathbb{PH} \otimes I_3)\boldsymbol{\phi}\|^2}{c_v\underline{\lambda}(\mathbb{Q})}
\end{aligned} \tag{23}$$

By solving (23), we get

$$\begin{aligned}
V_o(t) &\le V_o(0)\exp\left(-\frac{1}{2}c_0\frac{\underline{\lambda}(\mathbb{Q})}{\bar{\lambda}(\mathbb{P})}t\right) + \frac{2\bar{\lambda}(\mathbb{P})\|(\mathbb{PH} \otimes I_3)\boldsymbol{\phi}\|^2}{c_v^2\underline{\lambda}^2(\mathbb{Q})} \\
&\times \left(1 - \exp\left(-\frac{1}{2}c_0\frac{\underline{\lambda}(\mathbb{Q})}{\bar{\lambda}(\mathbb{P})}t\right)\right)
\end{aligned} \tag{24}$$

Recalling the definition of $V_o(t)$, it follows from (24) that

$$\|\bar{\varepsilon}\| \le \varrho \triangleq \max\left\{ \sqrt{\frac{\bar{\lambda}(\mathbb{P})}{\underline{\lambda}(\mathbb{P})}}\frac{2\|(\mathbb{PH} \otimes I_3)\boldsymbol{\phi}\|}{c_v\underline{\lambda}(\mathbb{Q})}, \sqrt{\frac{\bar{\lambda}(\mathbb{P})}{\underline{\lambda}(\mathbb{P})}}\|\bar{\varepsilon}(0)\| \right\}. \tag{25}$$

Then, from (15) and the definition of $\bar{e}_l$, it is deduced that $\|e_l\| \le \|\bar{e}\| \le \varrho/\underline{\sigma}(\mathbb{H})$, where $\underline{\sigma}(\mathbb{H})$ stands for the minimum singular value of $\mathbb{H}$. As a result, it can be concluded that the formation errors $e_l$ ($l \in \mathbb{V}$) converge to a residual set, which can be made arbitrarily small by increasing $c_v$.

Next, we prove that Zeno behavior can be strictly precluded. Define $\beta_k(t) = \|\tilde{h}_k(t)\|^2$. From the boundedness of the boundedness of $v_{l_1,1}, v_{l_1,2}$ and $\bar{v}_l$ ($l_1 \in \mathbb{V}_0, l \in \mathbb{V}$), it can be checked that $\tilde{h}_k(t)$ and $\dot{h}_k(t)$ are bounded. As a result, there exists a constant $\bar{\beta}_k > 0$ such that

$$\frac{d}{dt}|\beta_k(t)| = |2\tilde{h}_k^\top(t)\dot{h}_k(t)| \le \bar{\beta}_k, \forall t \in (t_{k,\mu}, t_{k,\mu+1}). \tag{26}$$

On the other hand, according to (10), we have

$$\beta_k(t_{k,\mu}) = 0, \quad \lim_{t \to t_{k,\mu+1}^-} \beta_k(t) = \frac{\psi_k^2}{c_v^2}, \tag{27}$$

which together with (26) gives $t_{k,\mu+1} - t_{k,\mu} \ge \psi_k^2/(c_v^2\bar{\beta}_k)$. Hence, Zeno behavior is strictly precluded, which completes the proof. ∎

## B. Formation Controller Design

Based on the obtained observer state $v_{l,1}(t)$, we aim to force $\boldsymbol{\xi}_l$ to track $v_{l,1}(t)$, $l \in \mathbb{V}$. Note that all variables related to the altitude of UGVs are set as zero. Therefore, we define the error as $z_{l,1} = \boldsymbol{\xi}_l - v_{l,1}$. First, consider the Lyapunov function candidate as $V_{l,1} = 0.5z_{l,1}^\top z_{l,1}$, it follows from (2), (5), and (7) that

$$\begin{aligned}
\dot{V}_{l,1} &= z_{l,1}^\top(\dot{\boldsymbol{\xi}}_{l,1} - v_{l,2}) \\
&= \begin{cases} z_{l,1}^\top(z_{l,2} + \boldsymbol{\alpha}_l - v_{l,2}), & l \in \mathbb{F}_1 \\ z_{l,1}^\top(G_l u_l + \boldsymbol{\vartheta}_l(t) - v_{l,2}), & l \in \mathbb{F}_2 \end{cases}
\end{aligned} \tag{28}$$

where $z_{l,2} = \dot{\boldsymbol{\xi}}_{l,1} - \boldsymbol{\alpha}_l$ with the virtual controller $\boldsymbol{\alpha}_l$ designed as follows:

$$\boldsymbol{\alpha}_l = -c_{l,1}z_{l,1} + v_{l,2} \tag{29}$$

where $c_{l,1} > 0$ is the design gain. The control input of $l$-th UGV is set as

$$u_l = G_l^{-1}(-c_{l,1}z_{l,1} + v_{l,2}) \tag{30}$$

where $G_l^{-1}$ exists due to $G_l$ is non-singular. Then, $\dot{V}_{l,1}$ is computed as

$$\dot{V}_{l,1} = \begin{cases} -c_{l,1}z_{l,1}^\top z_{l,1} + z_{l,1}^\top z_{l,2}, & l \in \mathbb{F}_1 \\ -c_{l,1}z_{l,1}^\top z_{l,1} + z_{l,1}^\top \boldsymbol{\vartheta}_l(t), & l \in \mathbb{F}_2 \end{cases} \tag{31}$$

In light of the Young's inequality, it holds that $z_{l,1}^\top \boldsymbol{\vartheta}_l(t) \le \frac{c_{l,1}}{2}z_{l,1}^\top z_{l,1} + \frac{1}{2c_{l,1}}\bar{\vartheta}_l^2$. Then, we have

$$\dot{V}_{l,1} \le -\frac{c_{l,1}}{2}z_{l,1}^\top z_{l,1} + \frac{\bar{\vartheta}_l^2}{2c_{l,1}}, \quad l \in \mathbb{F}_2 \tag{32}$$

To proceed, we consider the Lyapunov function candidate $V_{l,2} = V_{l,1} + 0.5z_{l,2}^\top z_{l,2}$, $l \in \mathbb{F}_1$, and we compute its derivative as

$$\begin{aligned}
\dot{V}_{l,2} &\le -c_{l,1}z_{l,1}^\top z_{l,1} + z_{l,1}^\top z_{l,2} + z_{l,2}^\top \dot{z}_{l,2}, \\
&= -c_{l,1}z_{l,1}^\top z_{l,1} + z_{l,1}^\top z_{l,2} + z_{l,2}^\top(m_l^{-1}u_l + \boldsymbol{\vartheta}_l(t) \\
&\quad - \dot{\boldsymbol{\alpha}}_l), \quad l \in \mathbb{F}_1
\end{aligned} \tag{33}$$

The control input of $l$-th UAV is set as

$$u_l = m_l(-c_{l,2}z_{l,2} + \dot{\boldsymbol{\alpha}}_l - z_{l,1}) \tag{34}$$

where $c_{l,2} > 0$ is the design gain and $\dot{\boldsymbol{\alpha}}_l = -c_{l,1}(\dot{\boldsymbol{\xi}}_l - v_{l,2}) + \bar{v}_l$. Then, we have

$$\dot{V}_{l,2} = -c_{l,1}z_{l,1}^\top z_{l,1} - c_{l,2}z_{l,2}^\top z_{l,2} + z_{l,2}^\top \boldsymbol{\vartheta}_l(t), \quad l \in \mathbb{F}_1 \tag{35}$$

Also, we have $z_{l,2}^\top \boldsymbol{\vartheta}_l(t) \le \frac{c_{l,2}}{2}z_{l,2}^\top z_{l,2} + \frac{1}{2c_{l,2}}\bar{\vartheta}_l^2$. Hence

$$\dot{V}_{l,2} = -c_{l,1}z_{l,1}^\top z_{l,1} - \frac{c_{l,2}}{2}z_{l,2}^\top z_{l,2} + \frac{\bar{\vartheta}_l^2}{2c_{l,2}}, \quad l \in \mathbb{F}_1 \tag{36}$$

It follows from (32) and (36) that

$$\begin{cases} \dot{V}_{l,2} \le -c_l V_{l,2} + \dfrac{\bar{\vartheta}_l^2}{2c_{l,2}}, & l \in \mathbb{F}_1 \\[2mm] \dot{V}_{l,1} \le -c_{l,1}V_{l,1} + \dfrac{\bar{\vartheta}_l^2}{2c_{l,1}}, & l \in \mathbb{F}_2 \end{cases} \tag{37}$$

where $c_l = \min\{2c_{l,1}, c_{l,2}\}$.

*Theorem 1:* Under Assumptions 1–3, consider the auxiliary filters with the event-triggered mechanism (10), and the virtual controller (29) and the control inputs (30) and (34), therefore, we derive the following outcomes.

1) All the signals remain bounded.
2) The formation tracking error $\boldsymbol{\epsilon}_l = \boldsymbol{\xi}_l(t) - \boldsymbol{\xi}_0(t) - \boldsymbol{o}_l$ eventually converges to a small neighborhood of origin.

*Proof:* Integrating (37) from 0 to $t$, it yields

$$
\begin{cases}
V_{l,2}(t) \leq V_{l,1}(0)e^{-c_l t} + \dfrac{\bar{\vartheta}_l^2}{2c_l c_{l,2}}(1 - e^{-c_l t}), & l \in \mathbb{F}_1 \\
V_{l,1}(t) \leq V_{l,1}(0)e^{-c_{l,1} t} + \dfrac{\bar{\vartheta}_l^2}{2c_{l,1}^2}(1 - e^{-c_{l,1} t}), & l \in \mathbb{F}_2
\end{cases}
\tag{38}
$$

Hence

$$
\begin{cases}
\|\boldsymbol{z}_{l,1}(t)\| \leq \dfrac{\bar{\vartheta}_l}{\sqrt{c_l c_{l,2}}}, & l \in \mathbb{F}_1 \\
\|\boldsymbol{z}_{l,1}(t)\| \leq \dfrac{\bar{\vartheta}_l}{c_{l,1}}, & l \in \mathbb{F}_2
\end{cases}
\tag{39}
$$

as $t \to \infty$. Thus, $\boldsymbol{z}_{l,1}$ is bounded. Based on similar procedure, $\|\boldsymbol{z}_{l,2}\|$ and $\boldsymbol{\alpha}_l$ are also bounded. Since $\boldsymbol{z}_{l,2} = \dot{\boldsymbol{\xi}}_{l,1} - \boldsymbol{\alpha}_l$, $\dot{\boldsymbol{\xi}}_{l,1}$ is bounded. It is deduced that all the signals remain bounded. Define $\sigma_{l,1} = \max\{\frac{\bar{\vartheta}_l}{\sqrt{c_l c_{l,2}}}, \frac{\bar{\vartheta}_l}{c_{l,1}}\}$, we have $\|\boldsymbol{z}_{l,1}(t)\| \leq \sigma_{l,1}$. Since $\boldsymbol{\epsilon}_l = \boldsymbol{\xi}_l(t) - \boldsymbol{\xi}_0(t) - \boldsymbol{o}_l$, $\boldsymbol{z}_{l,1} = \boldsymbol{\xi}_l - \boldsymbol{v}_{l,1}$, and $\boldsymbol{e}_l = \boldsymbol{v}_{l,1} - \boldsymbol{\xi}_0 - \boldsymbol{o}_l$, it yields from Lemma 2 that $\|\boldsymbol{\epsilon}_l\| \leq \varrho/\underline{\sigma}(\mathbb{H}) + \sigma_{l,1} := \rho$. As a result, the formation tracking error $\boldsymbol{\epsilon}_l$ eventually converges to a small neighborhood of origin. ∎

## IV. SIMULATIONS

In this stage, we will utilize two UAVs (labeled 1 and 4), two UGVs (labeled 2 and 3), and a leader (labeled 0), with the unknown disturbance denoted as $\boldsymbol{\vartheta}_l(t) = [0.1\sin(t); 0.1\cos(t); 0.2\cos(t)]$. The communication among these four agents and one leader is illustrated in Fig. 1.

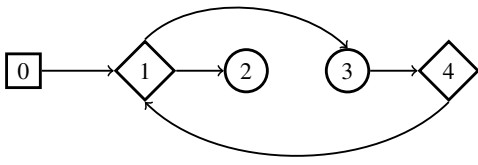

Fig. 1. Communication topology.

The leader's output is described as $\boldsymbol{\xi}_0(t) = [0.5\sin(t); 0.3t; 0.5t]$. The initial values are given as $\boldsymbol{\xi}_1(0) = [6; 0.5; 0]$ m, $\boldsymbol{\xi}_2(0) = [5; 0.1; 0]$ m, $\boldsymbol{\xi}_3(0) = [-4; 0; 0]$ m, $\boldsymbol{\xi}_4(0) = [-8; 0.3; 0]$ m, $\boldsymbol{v}_{1,1}(0) = \boldsymbol{\xi}_1(0)$, $\boldsymbol{v}_{2,1}(0) = \boldsymbol{\xi}_2(0)$, $\boldsymbol{v}_{3,1}(0) = \boldsymbol{\xi}_3(0)$, $\boldsymbol{v}_{4,1}(0) = \boldsymbol{\xi}_4(0)$, $\boldsymbol{v}_{0,1}(0) = \boldsymbol{\xi}_0(0)$, and the other initial values are set as zero/zero vector. The parameters are $\boldsymbol{o}_1 = [4; 0; 0]$, $\boldsymbol{o}_2 = [2; 0; 0]$, $\boldsymbol{o}_3 = [-2; 0; 0]$, $\boldsymbol{o}_4 = [-4; 0; 0]$, $m_1 = m_4 = 2$ kg, $p_l = 0.2$ m, $c_v = 50$, $\psi_k = 2$,

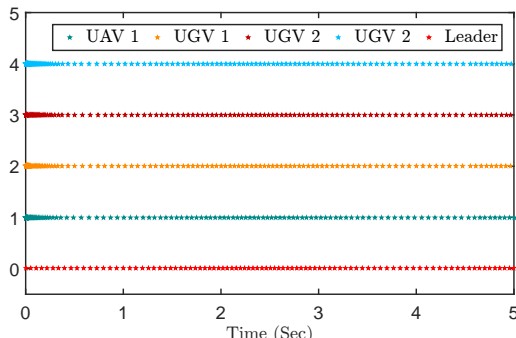

Fig. 2. Inter-event intervals of the leader, UAVs, and UGVs.

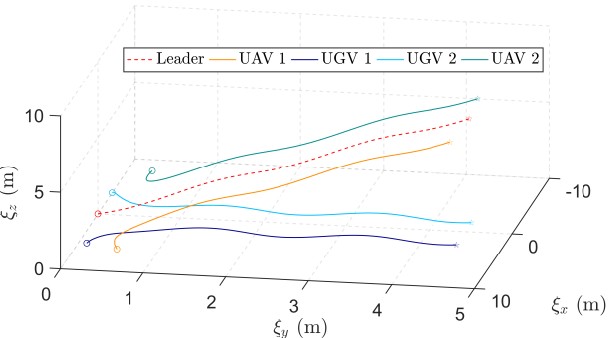

Fig. 3. Trajectories of $\boldsymbol{\xi}_l$, for $l \in \{0, 1, \ldots, 4\}$.

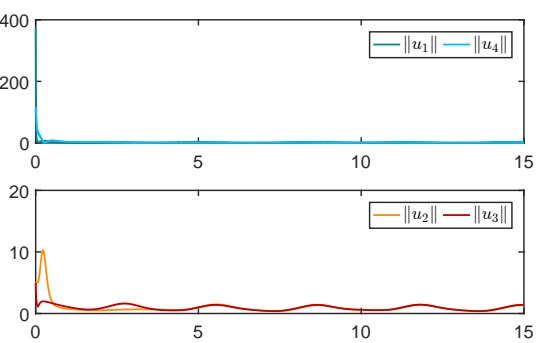

Fig. 4. Trajectories of $\|\boldsymbol{u}_l\|$, for $l \in \{1, \ldots, 4\}$.

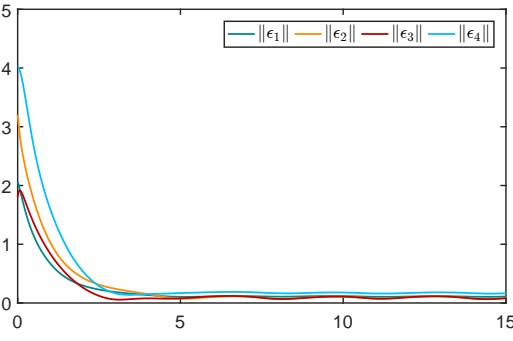

Fig. 5. Trajectories of $\|\boldsymbol{\epsilon}_l\|$, for $l \in \{1, \ldots, 4\}$.

TABLE I
COMPARISON OF TRIGGER NUMBERS.

| | Leader | UAV 1 | UGV 1 | UGV 2 | UAV 2 |
|---|---|---|---|---|---|
| Time-triggered | 15000 | 15000 | 15000 | 15000 | 15000 |
| Proposed observer | 283 | 360 | 371 | 336 | 382 |

$k = 0, 1, \ldots, 4$, $c_{1,1} = c_{4,1} = 5$, $c_{1,2} = c_{4,2} = 50$, and $c_{2,1} = c_{3,1} = 25$.

The simulation results are presented in TABLE I and Figs. 2–5. Fig. 2 shows that displays the inter-event intervals of the leader, UAVs, and UGVs, which presents an aperiodic form. As depicted in Fig. 3, all the UAVs and UGVs track the trajectory of the leader with formation configuration. In Fig. 4, the control inputs are bounded. From Fig. 5 it can be seen that the formation tracking errors converge to a small residual set in the presence of event-triggered communication. Thus, the effectiveness of the proposed method is verified.

## V. CONCLUSIONS

This paper investigates formation tracking for the UAV-UGV systems under event-triggered communications. A hierarchical control strategy is proposed, comprising a distributed event-triggered observer and a local controller, such that the formation tracking errors converge to a small residual set. The salient feature is that continuous communication among neighbors is not required, and the system exhibits wider robustness to disturbances. Finally, simulations validate its feasibility.

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
