# OpenReview forum: "Formation Tracking of UAV-UGV Systems Over Event-Triggered Communications"
_IEEE.org/ICIST/2024/Conference — IEEE ICIST 2024 Conference Submission_

### Official Review · Reviewer_gPww · 2024-08-21
**This paper investigates the formation tracking issue under event-triggered communications for unmanned aerial vehicle-unmanned ground vehicle systems with disturbances. A hierarchical control strategy is proposed, comprising a distributed event-triggered filter and a local controller. In my opinion, there are some comments that should be considered during the revision of the manuscript which are listed below.**

**Rating:** 7
**Confidence:** 3

**Review:**

1. It can be compared with existing articles to make the innovation point clearer.
2. The disadvantages of the proposed method and the direction of the next work should be explained in the conclusion.
3. In the simulation section, the explanatory text for each chart is insufficient, making it challenging to comprehend the author's intended message from the provided images. Please provide more detailed descriptions to enhance the reader's understanding.

---

### Official Review · Reviewer_fQZ5 · 2024-08-21
**accept**

**Rating:** 7
**Confidence:** 3

**Review:**

Comment: This paper proposes a hierarchical control strategy is proposed, comprising a distributed event-triggered filter and a local controller for the formation tracking issue under event-triggered communications. The theory is correct and can be accepted after responding the following comments.
(1) More comprehensive literature review is needed to clarify the research gap and research motivation.
(2) In the simulation section, more analysis can be added to better explain the main results of this paper, that's not enough.
(3) In the end of the conclusions, some research directions are suggested to be added.

---

### Official Review · Reviewer_GRc4 · 2024-08-22
**This article is very interesting and a good one**

**Rating:** 7
**Confidence:** 3

**Review:**

This paper proposed a hierarchical control strategy, comprising a distributed event-triggered filter and a local controller for UAV-UGVs. The obtained result is valuable and can be accepted if the following problems can be clarified.
(1) In the introduction, the shortages of those relevant studies are suggested to be further summarized.
(2) In the end of Section 1, the organization of this study is suggested to be summarized.
(3) In the simulation section, more analysis can be added to better explain the main results of this paper, that's not enough.
(4) The future work is missing in the Conclusion.
(5) There exist several spelling and grammar errors. Please check carefully and further polish

---

### Comment · Reviewer_GRc4 · 2024-08-21
**This article is very interesting and a good one**

This paper proposed a hierarchical control strategy, comprising a distributed event-triggered filter and a local controller for UAV-UGVs. The obtained result is valuable and can be accepted if the following problems can be clarified.
(1)	In the introduction, the shortages of those relevant studies are suggested to be further summarized.
(2)	In the end of Section 1, the organization of this study is suggested to be summarized.
(3)	In the simulation section, more analysis can be added to better explain the main results of this paper, that's not enough.
(4)	The future work is missing in the Conclusion.
(5)	There exist several spelling and grammar errors. Please check carefully and further polish

---

### Decision · Program_Chairs · 2024-09-06

Accept (Oral)